

# Scraping marking behaviour of the largest Neotropical felids

Francisco Palomares[1], Noa González-Borrajo[1], Cuauhtémoc Chávez[2], Yamel Rubio[3], Luciano M. Verdade[4], Rocio Monsa[4], Bart Harmsen[5], Begoña Adrados[1] and Marina Zanin[6]

[1] Department of Conservation Biology, Estación Biológica de Doñana, CSIC, Seville, Seville, España
[2] Universidad Autónoma Metropolitana-Unidad Lerma, Lerma, Estado de México, Mexico
[3] Escuela de Biología, Universidad Autónoma de Sinaloa, Sinaloa, Mexico
[4] Núcleo Laboratorial de Ecologia Evolutiva Aplicada, Universidade de São Paulo, CENA, Piracicaba, Sao Paulo, Brazil
[5] Panthera, New York, United States of America
[6] Departamento de Biologia, Universidade Federal do Espírito Santo, Vitória, Espírito Santo, Brazil

## ABSTRACT

**Background**. Details of how, why and in what conditions large felids make scrapes is unknown. Here, we examined the general hypothesis about the use of scrapes for marking proposals, as well as to communicate with other individuals to signalize particular points or areas of interest, by studying scrape-marking behaviour of jaguars and pumas.

**Methods**. We surveyed by scrapes between five days and two months mainly during dry season in five study areas from Mexico (El Edén and San Ignacio), Belize (Cockscomb) and Brazil (Angatuba and Serra das Almas), which differed in presence and/or abundance of jaguars and pumas. Paths were slowly walked while searching for scrapes by teams normally composed of two people and tracks were stored in GPS, distinguishing the type of path surveyed (unpaved track roads, trails and cross-country).

**Results**. We found a total of 269 felid scrapes along 467 km of paths surveyed, obtaining a finding rate of 0.576 scrapes per km. Most scrapes were found in car tracks (0.629 scrapes per km), followed by trails (0.581 scrapes per km), and rarely did we find scrapes in cross country (0.094 scrapes per km). In trails, scrapes were found in a similar frequency in the centre and edge, whereas in car tracks they were mainly found in the edge. There were also clear differences in the position of the scrapes between study areas that differed in presence and/or abundance of pumas and jaguars, with scrapes located mainly in the centre in areas only with pumas, in the centre and in the edge in areas with a similar number of jaguars and pumas, and in the edge in area mainly dominated by jaguars. The remarking rate tended to be higher in one of the areas with only pumas where natural vegetation was scarcer. Felids chose sites mainly covered by leaves and located in paths less wide, clean and rarely used.

**Discussion**. Scraping was a frequent behaviour in the largest felids of America, although in some areas, scraping behaviour was rare. Scrapes seem to be signalizing some specific areas within territories and data suggest that they are made with the proposal of communication between individuals. It seems that a high scraping behaviour in pumas is not related to the presence of jaguars.

Corresponding author
Francisco Palomares,
ffpaloma@ebd.csic.es

## INTRODUCTION

Communication plays an important role in mammalian populations and communities, providing, for instance, information about the presence, identity, health or social status of individuals of the same or different species, or location of feeding places within territories and their limits (*Johnson, 1973*; *Mellen, 1993*; *Gosling & Roberts, 2001*). Mammals use visual, tactile, vocal or olfactory signals to communicate to each other or to signalize the area (*Reiger, 1979*; *MacDonald, 1980*; *Gorman & Trowbridge, 1989*), and particularly in carnivore mammals, scent marking using urine, secretions, ground scratching, rubbing, or faeces has been well-documented (*Asa, Mech & Seal, 1985*; *Smith, McDougal & Miquelle, 1989*; *Barja, De Miguel & Bárcena, 2005*). Solitary carnivores have spatially dispersed populations and indirect scent-marking signals are the most frequently via of intraspecific and interspecific communication. Among the types of indirect marking signals, scratches present the characteristics of functioning both as visual and olfactory signals, since the ground is removed and odour is added from the paw glands (*Schaller, 1972*; *Peters & Mech, 1975*; *Bekoff, 1979*).

Scraping or scratching on the ground is a known behaviour in large felids (*Emmons, 1987*; *Jackson & Ahlborn, 1988*; *Ghoddousi et al., 2008*). However, detailed description of this marking behaviour, how and in what conditions felids make scrapes, and the intention behind felid scraping behaviour, are largely unknown and untested in most species and situations (but see *Smith, McDougal & Miquelle, 1989*; *Harmsen et al., 2010a*; *Allen, Wittmer & Wilmers, 2014*). Scrapes are depressions in the ground made by felids with their hind legs, moving back the dirt and making a pile of it at the end of the scrape (*Smith, McDougal & Miquelle, 1989*; *Harmsen et al., 2010a*; *Logan & Sweanor, 2010*; Fig. 1), although in occasions the front leg may also be used (*Harmsen et al., 2010a*).

Scrapes seem to be regularly created along territorial boundaries or prominent travel-ways (*Seidensticker et al., 1973*; *Logan & Sweanor, 2010*; *Harmsen et al., 2010a*). At least in pumas, most scrapes are thought to be made primarily by mature males and less often or not at all by mature females or inmature individuals (*Logan & Sweanor, 2010*; *Harmsen et al., 2010a*; *Allen, Wittmer & Wilmers, 2014*). *Harmsen et al. (2010a)* specifically studied scrape-marking behaviour in jaguars and pumas in one area of Central America, and found that both species scrape, although scrapes were spatially clustered along trails and male pumas scraped more often. Moreover, the scrapes tended to be larger in jaguars than in pumas, although not enough to be distinguished only by size.

Here, we studied scrapes and the scrape-marking behaviour of jaguars and pumas in several ecologically different areas of the Neotropic. We chose for surveys areas with only pumas and other areas with the presence of both species but with different densities of each felid in order to be able to ascertain a possible interaction between the scrapes morphology and scrape-marking behaviour and the sympatry between both species. We specifically aimed first to describe how scrapes are, and the characteristic of the places where, scrapes

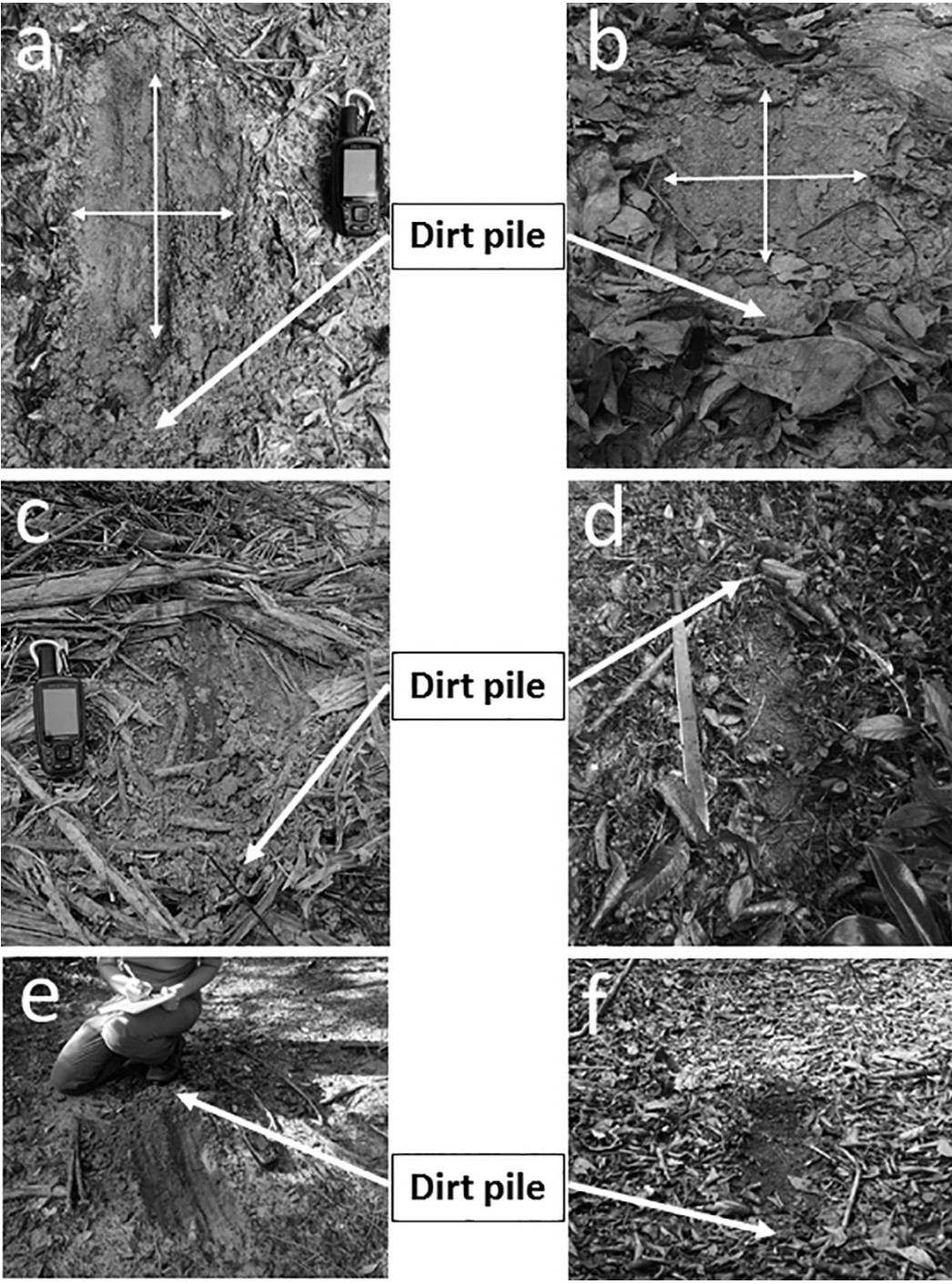

**Figure 1  Types of large felid scrapes.** Examples of scrapes with different sizes and on different substrates. (A–C) are puma scrapes from Angatuba and the other ones could be from pumas or jaguars, although (D) and (F) could be from jaguars since they are much longer than wider and no marks from the two legs are observed (see text for details). Mark of claws of both legs (even back leg foot tracks) are clearly appreciated in (A) and (E) scrapes. The pile of dirt on the end of the scrape is marked in all cases, and in (A) and (B) length and width were measures indicated as well (note than (B) is one case where length and width was similar). Photos by Francisco Palomares.

are found, examining if there are differences in scrapes and the scraping behaviour between areas differing in presence and density of both felids. Secondly, we examined the general hypothesis about both felids species using scrapes for marking proposals, as well as to communicate with other individuals and to signalize particular points or areas of interest. If so, we expect (1) they will select specific points with particular characteristics which would facilitate other individuals detecting them, (2) the area around the point would present some distinctive general habitat characteristics in relation to other parts of the study area, and (3) scrapes will not be evenly distributed in space and will be concentrated in some given points where re-scraping will be frequent.

## MATERIAL & METHODS

### Study areas

We carried out the study mainly in five areas (Fig. 2): (1) two zones around San Ignacio municipality (México), (2) El Edén Ecological Reserve (México), (3) Cockscomb Basin Forest Reserve (Belize), (4) three sites at Angatuba municipality (Brazil), and (5) the Serra das Almas Private Reserve of Natural Heritage (Brazil). The surveyed areas in San Ignacio (23°31′ and 24°26′N, 105°44′ and 106°44′W) have sandy and clay soils, and are dominated by deciduous and semideciduous tropical forest with some areas of dry tropical forest (*Rubio, Bárcenas & Beltrán, 2010*). El Edén (21°130′N, 87°110′W), located in the State of Quintana Roo, México, mainly has rocky soils and includes a great variety of ecosystems, mainly formed by semideciduous dry tall tropical forest dominated by trees (up to 25 m), and tropical short forest (up to 12 m) in the lower part, which usually floods during the rainy season, savanna-like vegetation with some palm trees also floods during the rainy season, and secondary regeneration plant communities located in formerly agricultural or cattle areas (*Torres-Barragán et al., 2004*). Cockscomb (16°47′N, 88°37′W) has sandy and clay soils and was heavily logged until the 1980s; in 1990 it was declared a wildlife sanctuary. Thus, the vegetation of Cocskomb is composed mainly of well-developed secondary, moist, broadleaf tropical forest at several stages of succession (*Harmsen et al., 2010a*; *Rabinowitz & Nottingham, 1986*). Angatuba is located in the south-central region of São Paulo state, and surveys were carried out in two adjacent ranches (Três Lagoas, 23°22′S, 48°28′W, and Arca, 23°20′S, 48°27′W), and in the Angatuba Ecological Station (23°24′S, 48°21′W), located 9 km from the ranches. Angatuba has sandy soils, a humid subtropical climate, is situated in the transitional region between the Atlantic Forest and Cerrado biomes, and some patches of autochthonous (Cerrado and Atlantic Forest) vegetation remain (*Athayde et al., 2015*), although most of the area in the ranches, and in lesser extension in the Ecological Station, has been forested with eucalyptus and pine plantations. Serra das Almas is a private protected area located at the Ceará State in the Brazilian Caatinga biome (5°8′29.15″S; 40°54′58.60″W). Three main phytophysiognomies of Caatinga biome are found in this reserve: seasonal and dense shrub vegetation, seasonal deciduous thorn forest, and montane seasonal deciduous forest (*Araújo & Martins, 1999*; *Lima et al., 2009*).

Pumas were the only large felid species present in Serra das Almas and Angatuba, whereas both species, pumas and jaguars, were present in the other three study areas, although in

Peer J ___________________________________________

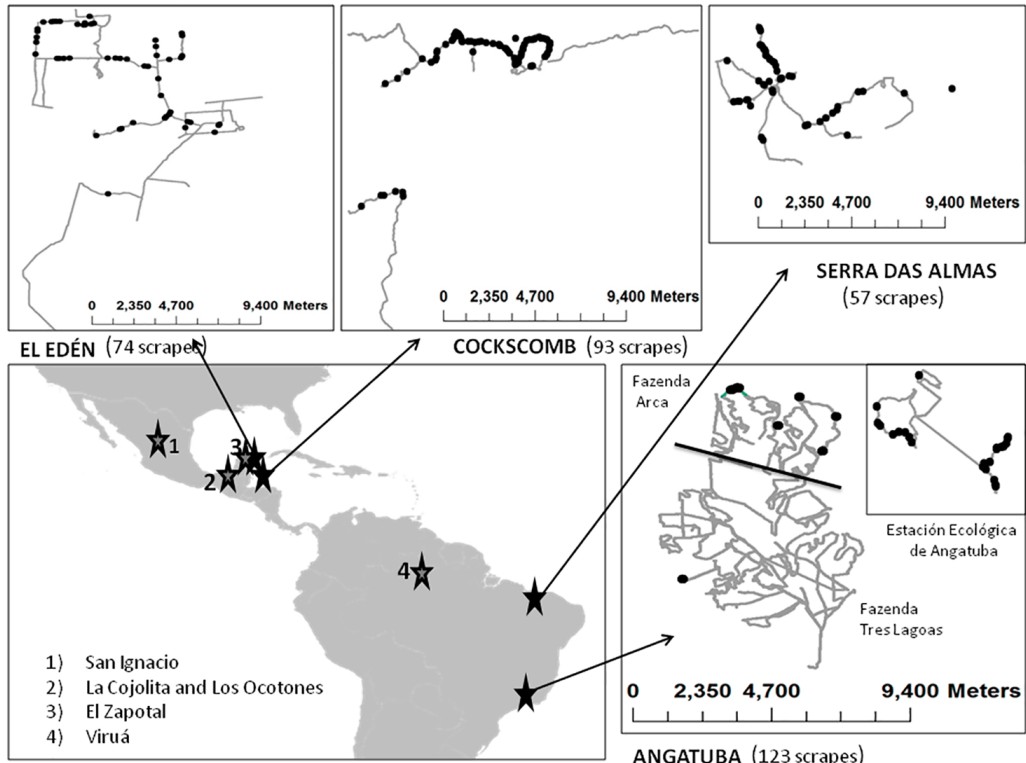

**Figure 2** **Main study areas and surveys.** Sketch of the principal study areas (Angatuba, Cockscomb, El Edén and Serra das Almas), tracks surveyed (lines) and total scrapes found (points). Details of San Ignacio study area is not shown since only two scrapes were found, and on the general map the location of other areas where we obtained extra information used in this study are indicated with numbers (see 'Felid relative abundance and assigning scrapes to species and sex' section).

one of them (Cockscomb) jaguars are much more abundant than pumas (*Harmsen et al., 2010b*, and see below).

We also analysed the faeces or urine we found over scrapes to try to determine species making scrapes (see below), and for this analysis, in addition to the data gathered in our main study areas, we also included other cases obtained along surveys in other areas from Latin-America such as the Viruá National Park in the Amazon basin (Brazil), the Ecological Reserve El Zapotal in Quintana Roo (México), and in Los Ocotones and La Cojolita in Chiapas (México; Fig. 2; details of these surveys and areas in *Palomares et al., 2012*; *Palomares et al., 2016*).

## Surveying for scrapes

Surveys lasted between five days and two months, and were mainly carried out during the dry season in all study areas, except in El Edén and Cockscomb when samplings also included the transition with the rainy period (in April 2012 in San Ignacio, in September–October 2012 in Angatuba, in May–June 2013 in El Edén, in April–June 2014 in Cockscomb, and one in September and another one in November 2015 in Serra das Almas). Paths were

only surveyed once except in Serra das Almas, which was sampled twice as mentioned). Nevertheless, in El Edén and Cockscomb, we also recorded opportunistically some other scrapes after the first survey since we continued with other activities, which were not considered by estimating finding rates, but they were considered for estimating scrapes size, position in the paths, or re-scraping rate in order to not lose biological information. All paths were slowly walked (about 2–3 km per hour) while searching for scrapes by teams normally composed of two people and tracks stored in GPS, distinguishing the type of path surveyed (unpaved track roads, trails and cross-country). We took attention in looking for scrapes within path limits and in their borders, so when we detected felid tracks leaving the paths we tracked them when possible outside of the path in order to detect additional scrapes outside from it. This was particularly possible in the study areas with sandy substrates such as Angatuba and Serra das Almas.

Most times a scrape was found, their position was recorded with a GPS, along other associated information such as position in the path (centre, wheel tracks, border, outside and faraway; see below), type of substrate (ground or rock), cover of substrate (clean, grass, woody material, leaves), type, height and cover of vegetation around (in a 25 m diameter circle centred in the scrape), the use of the path by people (habitual, rare or abandoned), and if the path had any kind of maintenance or not. We did not record if scrapes had single or double rakings as described by *Harmsen et al. 2010a* (Fig. 1). Scrapes were considered to be in: (1) centre when situated in the space between the wheel tracks for the case of unpaved track roads, or within the walkable part in the case of trails; (2) wheel tracks when just in the tracks left for wheels for the case of unpaved road; (3) border, outside or faraway when it was up to 1 m, between 1–5 m, and >5 m outside of the workable part of trails or from the wheel tracks of unpaved roads.

Additionally, when scrapes were relatively recent and, therefore, size was clearly defined, their length (without the pile of dirt accumulated in the back of scrape) and width were measured (Fig. 1). Scrape length was taken at the direction in which the scrape was made by the felids, which could most times easily be distinguished by tracts left by the fingers of animals, and if not by the accumulated dirt on one of the extremes (Fig. 1).

Sampling in Brazil was carried out under licenses no. 11214 and no. 13781 of ICMBio, and no 131/2005 CGFAU/LIC, 13883-1 SISBIO and 15664-1 SISBIO of the Instituto Brasileiro do Meio Ambiente—IBAMA, and at the Mexican sites under the licence SGPA/DGVS/549 of the Dirección General de Vida Silvestre (Semarnat). Sampling in Belize and posterior exportation of the faeces was approved by the Forest Department of Belize with the Ref. No. CD/60/3/14 (25). Faecal samples were exported from Brazil to Spain for genetic analysis under IBAMA/CGEN Autorização de Acesso licence no 063/05 and IBAMA/CITES export licences no 0123242BR, 08BR002056/DF and 09BR003006/DF, and from Mexico to Spain under the export licences no MX33790 and MX42916 of the Secretaria de Medio Ambiente/CITES.

## Selection of scraping sites

We examined if there was any kind of selection of the points used for scraping at two different scales. First, we tested the selection at the microsite scale of the scrape (i.e.,

substrate type and ground cover), in a plot of 1 × 1 m side situated around the scrape, and compared it with another 1 × 1 m plot situated just 5 m further on the same path walked. The distance of 5 m was arbitrarily chosen on the basis of being far enough to present a random sampling of the characteristics of the ground in the path, but close enough to be sure conditions of the path were similar. Within each 1 m plot we estimated the percentage of clean ground or ground covered by grass, leaves, rocks, or woody material. Second, we looked if the path and vegetation characteristics of the scrapes presented any particular features (path use, width and maintenance, and vegetation height, cover and type). These data were compared with their availability in the study area, which was obtained by sampling these same variables every 500 m on the paths surveyed for scrapes. The vegetation types considered for each study area were Atlantic Forest, Cerrado, and plantations (eucalypt and pine altogether) in Angatuba; savannas, tall forest, short forest and secondary forest (regeneration forest) in El Edén; and tall forest and secondary forest in Cockscomb. In many occasions the paths were located just in the joint border of two vegetation types, so we independently recorded vegetation in each side of the path, and considered it as edge for analysis. Vegetation characteristics were measured in a 25 m diameter circle centred in the scrape.

## Felid relative abundance and assigning scrapes to species and sex

We got an index of jaguar and puma relative abundance (ratio jaguar/puma faeces) in each study area by collecting faeces in the same tracks where scrapes were recorded.

On the other hand, when we found faeces or urine over scrapes we collected a piece of the faeces or the leaves with urine in order to identify the species and the sex of individual that deposited them.

Faeces and urine were analyzed by non-invasive molecular techniques previously developed in our lab (see *Roques et al., 2011*; *Roques et al., 2014* for further details). Although we cannot be sure the faeces or urine found over scrapes necessarily had to come from the species and individuals who made the scrape, we considered it relevant to include this information. As mentioned previously in the Study Area section, for this analysis, in addition to the data gathered in our main study areas, we also included other cases obtained along surveys in other areas from Latin-America or in other samplings of the same areas.

## Data analysis

We could not get all measures and variables for all study areas, so for each analysis we will indicate for which areas data were available. Similarly, since there were many different types of data and comparisons, the statistical analysis performed will be mentioned in the appropriate part of the result section.

Linear distribution of scrapes on paths was described by determining first the number of scraping sites and the rate of re-scraping per site (i.e., the number of scrapes found in each site). A site was defined as an area where one or more scrapes were found within a given distance. To determine this distance, we used the maximum distance we found (21 m) between scrapes in two intensive scraping sites that we found in the Angatuba

area. Therefore, scrapes were considered as belonging to the same site if distance was equal or inferior to 21 m. We are aware this distance of 21 m might not be the same for all study areas and situations, but it is the only objective information we had about this issue. Furthermore, we also counted the number of scrapes we found between 0 and 21 m, between 22 and 100 m, between 101 and 500 m, and further than 501 m, in order to be able to recognize if there were different patterns of linear distribution of scrapes among study areas. These distance intervals were somewhat arbitrarily chosen, but according to the range of home range sizes of jaguars and pumas (see *González-Borrajo, López-Bao & Palomares, 2017*), it is reasonable to think that they represent scrapes made in the same site, in the same area, in a relatively close distance to another scrape, and in an isolated way, respectively.

Additionally, we determined if scrapes were spatially aggregated or randomly distributed on trails in each study area by the Ripley's reduced second moment function K(r) from a point pattern in a window and the translation correction (*Dixon, 2002*; *Ohser, 1983*). To adapt the Ripley's function to our specific case where scrapes were looked in linear transects instead of areas, we considered as window the effective searching area of each study site, which was formed by the total sampled transects with a width of 22 m. Therefore, the Ripley's function examines how distance between scrapes varied along the distance sampled in trails, and the curve formed by the Ripley's function may inform on the patterns of linear spatial distributional of scrapes on trails. To statistically test if the observed distribution of scrapes was significantly different from a random distribution, we generated 1,000 data sets of random positions of the same number, to that of observed scrapes in each study area, and looked to see if the observed Ripley's function overlapped or not with the expected ones. If the observed Ripley's function is above the expected ones it would indicate that scrapes were closer together than expected by random. The analysis was performed with the 'spatstat' package (*Baddeley & Turner, 2005*) in the R software (*R Core Team, 2016*).

# RESULTS

## Field samplings and finding rates

We walked 467 km of paths once or more times, finding a total of 349 felid scrapes, considering those found during surveys for scrapes (269) and those found while carrying out other activities (80). Considering for all areas only scrapes found during the specific samplings for scrapes (269 scrapes), we found a scraping rate of 0.576 scrapes per km surveyed (Table 1). The intensity of scraping behaviour differed among study areas and type of path. The highest number of scrapes was found in Angatuba, but once corrected by kilometers surveyed, Serra das Almas was the area where finding rate was higher, followed by El Edén, Angatuba, Cockscomb, and San Ignacio where only two scrapes were found in 67 km surveyed ($X^2 = 37.90$, *d.f.* $= 4$, $p < 0.001$; for differences among study areas; Table 1). Due to the low number of scrapes found in San Ignacio, we did not consider this area for further analysis. Scrapes were not found with the same probability in all types of path surveyed ($X^2 = 12.64$, *d.f.* $= 2$, $p = 0.002$). Most scrapes were found in car tracks (finding rate $= 0.629$ scrapes per km), followed by trails (0.581 scrapes per km), and rarely we found scrapes in cross country (0.094 scrapes per km; Table 1). Differences between car tracks and trails were not statistically significant ($X^2 = 0.17$, *d.f.* $= 1$, $p = 0.685$).
**Table 1  Scrape finding rates.** Kilometers surveyed and number of scrapes found in the five study areas and overall by type of path during the first survey in each study area except in Serra das Almas where data from two surveys were considered since the same track (37 km) was walked and time lap between surveys was 2 months.

| Study area | Type of path | | | | | | Overall | | |
|---|---|---|---|---|---|---|---|---|---|
| | Car tracks | | Trails | | Cross country | | | | |
| | Km | Scrapes | Km | Scrapes | Km | Scrapes | Km | Scrapes | Scrapes/km |
| San Ignacio | 26 | – | 17 | 1 | 24 | 1 | 67 | 2 | 0.030 |
| El Edén | 39 | 16 | 48 | 44 | 5 | 2 | 92 | 62 | 0.674 |
| Angatuba | 171 | 95 | 2 | 1 | 3 | – | 176 | 96 | 0.545 |
| Cockscomb | 33 | 11 | 25 | 16 | – | – | 58 | 27 | 0.466 |
| Serra Almas | 6 | – | 68 | 57 | – | – | 74 | 57 | 0.770 |
| Total | 275 | 173 | 160 | 93 | 32 | 3 | 467 | 269 | 0.576 |

## Position of scrapes in paths

Most scrapes were found in the border (43.7%) and centre (27.4%) of paths, although they also were recorded far away (15.7%), in the wheel car path (7.2%), and outside (6.0%) of the path ($n = 318$ scrapes with information). It is worth noting the relatively high number of scrapes found far away from paths, but data are biased because we found two high intensive scraping zones just 15–25 m from the edge of a car tracks and separated in between by 157 m in the Angatuba study area, where a total of 49 (39 plus 10, respectively) out of 96 scrapes found in the area were in these two points (Fig. 3). The two intensive scraping points covered a surface of 54 and 44 m$^2$, respectively, between their furthest scrapes and could be located after tracking a puma for approximately 800 m that was moving by the car tracks and entered this area (Fig. 3). This area was used during all the duration of the study and is even still used three years later when we visited the study area again. We set a camera track in the area and two days later we recorded a male puma making two consecutive scrapes with the hind legs (Supplemental Information). He was in scraping behaviour during 42 s and took 18 and 16 s in each scrape; he also deposited a small fecal substance after the first scrape. Three years later, we set another camera trap in the same point during four weeks, and we recorded four different pumas (two adult males, one adult female, and one individual of undetermined sex) visiting the point. Apart from these two intensive scraping sites, only one scrape was located far away from paths. For further analysis, these two intensive scraping sites will be considered as two data points if not otherwise indicated.

By type of path, considering data from all study areas altogether, scrapes were found in a similar frequency in the centre and edge of trails, whereas they were mainly found in the edge in car tracks (Fig. 4). But the most interesting result was that there were clear differences between study areas (trails: $X^2 = 45.02, d.f. = 2, p < 0.001$; car tracks: $X^2 = 37.02, d.f. = 6, p < 0.001$; data from edge and outside grouped for analysis in both cases; Fig. 4). In the areas with only pumas (Angatuba and Serra das Almas), scrapes were mainly found in the centre of the paths when these were trails, and in the wheel tracks when paths were car tracks. However, in the areas with jaguars and pumas, most scrapes

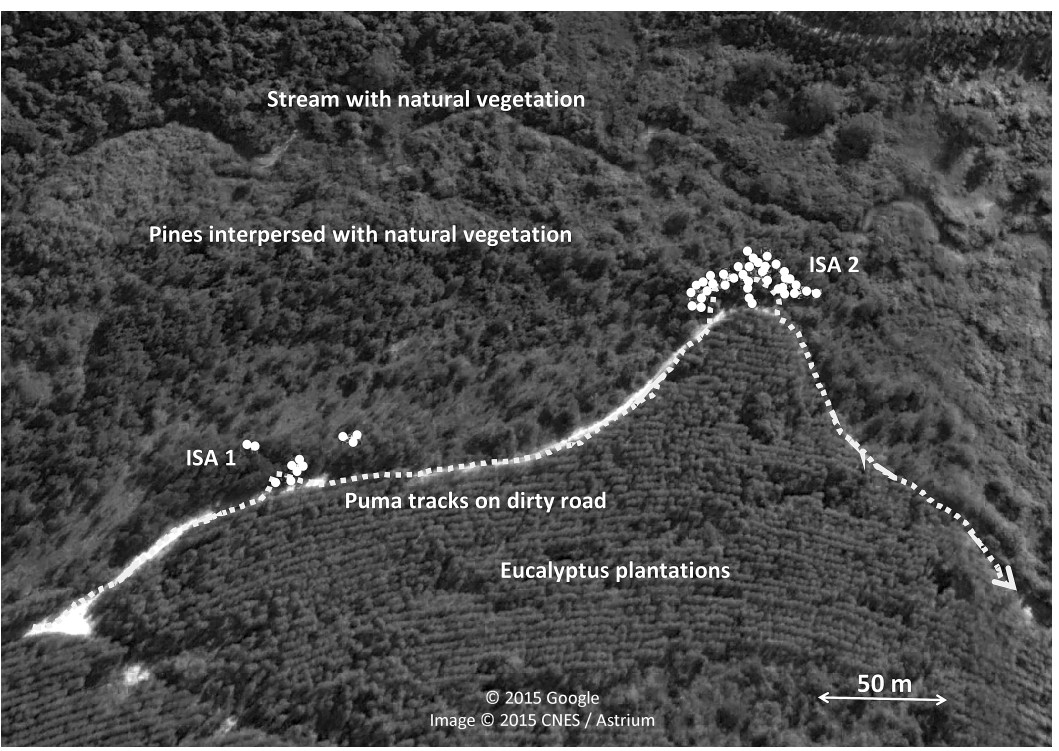

**Figure 3 Intensive scraping points of pumas.** Sketch of the two intensive scraping points (ISA1 and ISA2) found after following a puma track in Angatuba study area during the survey of 2012. Aerial images obtained from Google Earth.

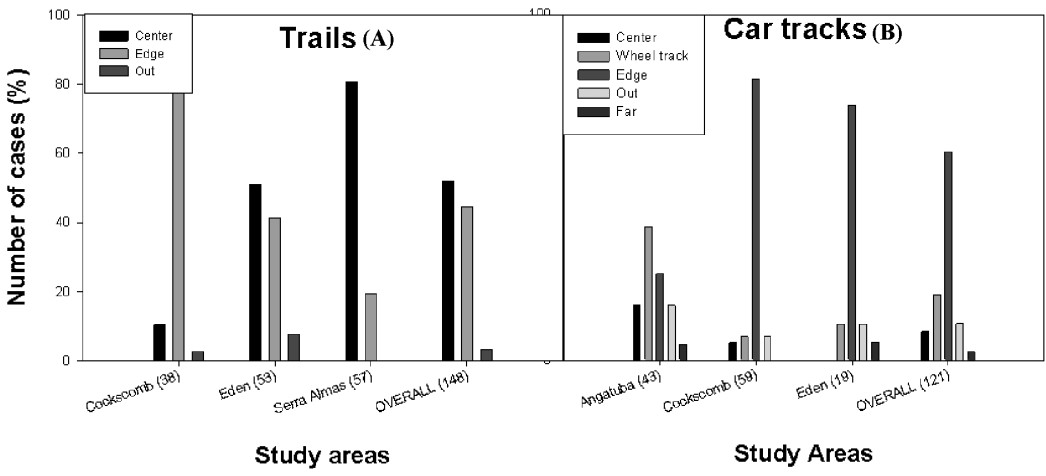

**Figure 4 Position of scrapes on paths.** Frequency distribution of the position of the scrapes on trails (A) and car tracks (B) in Angatuba, Cockscomb, El Edén, Serra das Almas and overall. There were no scrapes on trails in Angatuba and on car tracks in Serra das Almas.

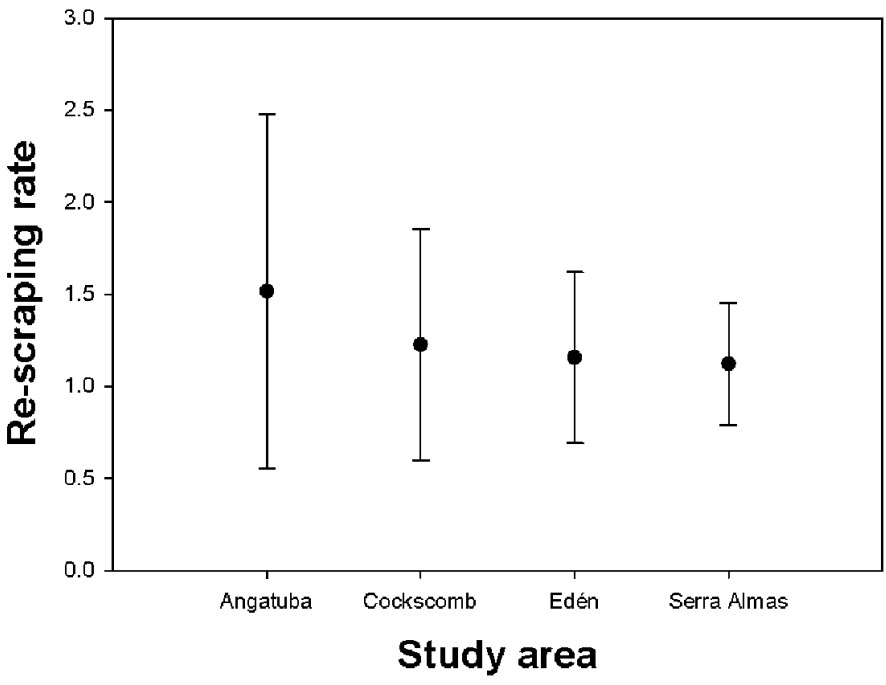

**Figure 5  Re-scraping behaviour.** Mean and standard deviation of the number of scrapes found on the same sites (i.e., <22 m between scrapes) in each study area.

were found in the centre or edge (El Edén), or in the edge (Cockscomb) when trails, and in the edge when car tracks in both areas (Fig. 4).

### Re-marking rate and linear distribution of scrapes on paths

We recorded a total of 347 scrapes located in 207 scraping sites (i.e., <22 m far from each other), with a mean of 1.44 (SD = 2.63, range = 1–39) scrapes per site. If we removed from analysis the two intensive scraping sites found in Angatuba, the mean number of scrapes per site was 1.23 (SD = 0.61, range = 1–5), and the Kruskal-Wallis test suggested differences between study areas ($H = 7.375, d.f. = 3, p = 0.061$), tending the number of scrapes per site to be slightly higher in Angatuba than in the other three areas (Fig. 5).

The representation of the Ripley's function showed a similar pattern in scrape distribution on trails between El Edén and Serra das Almas, and different ones in Angatuba and Cockscomb, but in all cases scrapes were clearly aggregated on trails, except in the Angatuba area when they were found according a random distribution when distance between scrapes was higher than 3000 m (Fig. 6). Overall, about 40% of scrapes were found in distances between 100 and 500 m far from each other, whereas scrapes very close (<22 m), in the same areas (between 22–100 m), or separated by >500 m were found in similar frequencies (around 20%; Fig. 7). However, the chi-square test detected significant differences among study areas ($X^2 = 33.20, d.f. = 9, P < 0.001$), and in Angatuba, scrapes were more frequently found in close proximity (i.e., re-scraping behaviour in the same site was high; data from the two intensive scraping sites removed for analysis), and in

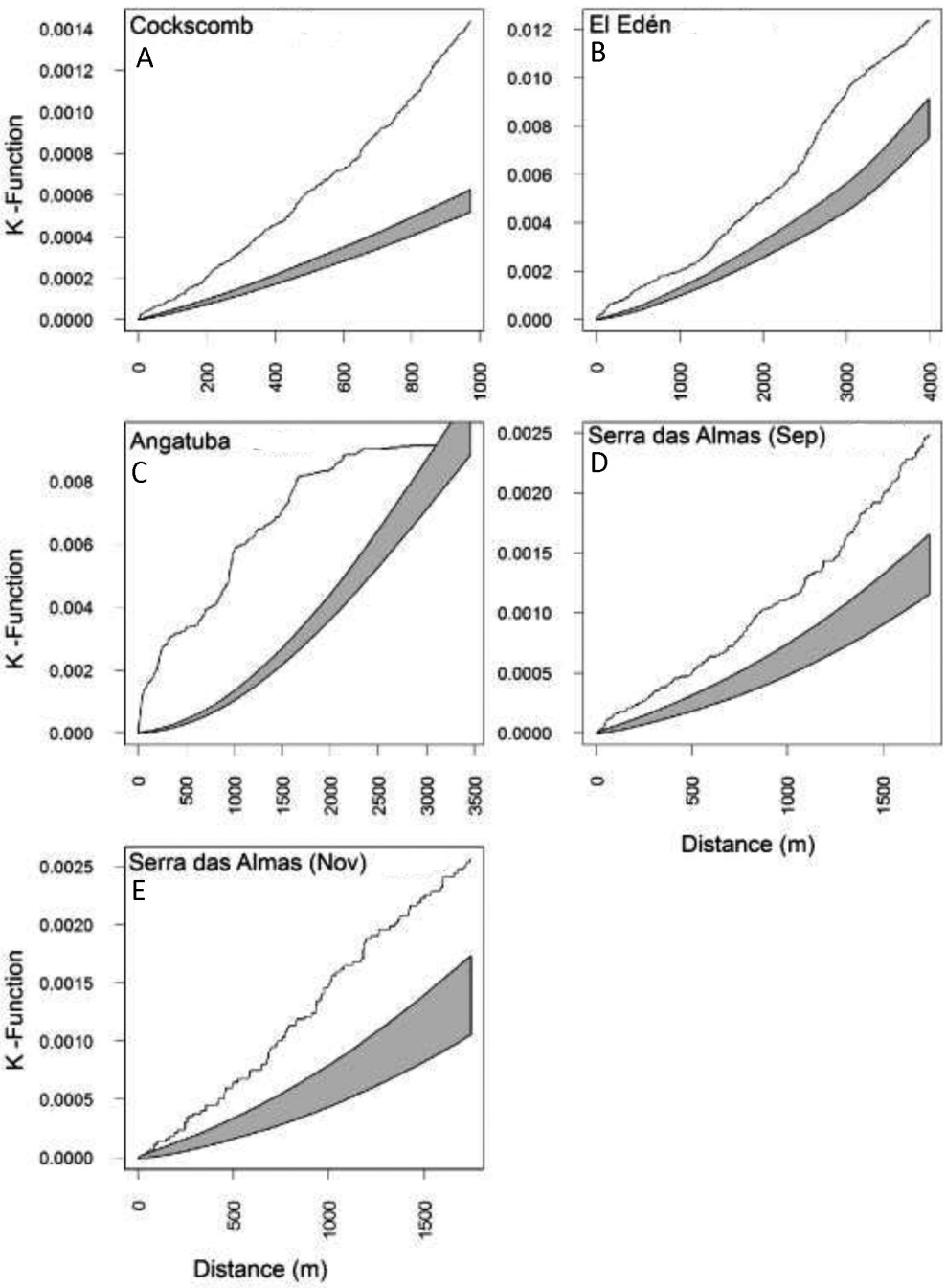

**Figure 6 Lineal distribution of scrapes on paths.** Estimates of the Ripley's reduced second moment function K(r), with the translation correction of *Ohser (1983)*, from a point pattern of scrapes in trails sampled in each study area (A–E). The gray shadow represents the expected distribution of scrapes following 1,000 random distributions (see text for details). The black line show the observed distribution of distances between consecutive scrapes. When the line is above the shadow area indicates that scrapes are closer together than expected by random.

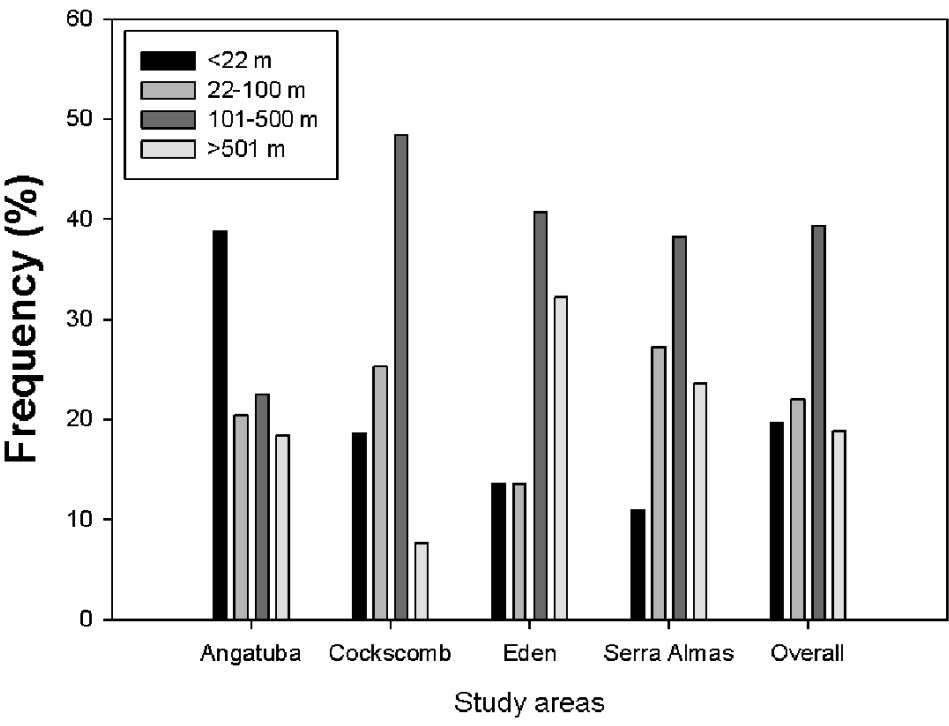

**Figure 7** **Lineal distance between consecutive scrapes.** Frequency distribution of distances between scrapes on linear paths for all data altogether and for every study area. There were significant differences between study areas ($X^2 = 33.202, d.f. = 9, P < 0.001$), and also between all pair wise comparisons (all $X^2 > 8.53, d.f. = 3$, all $P < 0.036$), except between El Edén and Serra das Almas ($X^2 = 3.605, d.f. = 3, P = 0.307$).

Cockscomb, felids made scrapes every 100–500 m with some re-scraping behaviour along transects, but also large sections (>500 m) of paths without scrapes (Fig. 7). Linear distribution of scrapes in El Edén and Serra das Almas, however, was not statistically different (Fig. 7), with scrapes mainly found every 100–500 m, although there also were large sections (>500 m) of paths without scrapes.

## Characteristic of sites with scrapes

Felids made scrapes on sites mainly covered by leaves (88 cases; Fig. 1), and in less extension on sites with grass (five cases), no coverage (three cases) or woody material (one case) in the area of Cockscomb, and on sites mainly covered by leaves (58 cases), but also with no coverage (11 cases) or woody material (one case) in El Edén. When we compared the materials covering the 1 × 1 m plots where scrapes were made, with the paired 1 × 1 m plots situated 5 m away from scrapes to test for any kind of selection, we found that whereas in Cockscomb felids did not show any kind of selection, in El Edén they preferred to make scrapes in sites with leaves and without stones and grass (Table 2).

For all study areas with data, felids preferred scraping paths less wide (between 1.7 and 3.0 m, depending of the study area), clean paths and rarely used (except for Cockscomb, where they preferred frequently used paths) than available; abandoned paths were used

**Table 2 Characteristics of microsites with scrapes.** Mean and standard deviation of percentages of the soil covered by different materials in 1 × 1 m plots centred in scrapes and 5 m away in front of the scrape on the path in the study areas of Cockscomb and El Edén. Probabilities of the Mann–Whitney $U$ test statistic to compare the paired 1 × 1 m plots are also showed.

| Materials | Cockscomb ($n = 93$) | | | El Edén ($n = 65$) | | |
|---|---|---|---|---|---|---|
| | Mean ± SD | | $P$ | Mean ± SD | | $P$ |
| | Scrape | 5-m-away | | Scrape | 5-m-away | |
| Nothing | 11.8 ± 21.36 | 7.0 ± 17.43 | 0.092 | 11.3 ± 22.56 | 18.4 ± 29.14 | 0.242 |
| Grass | 2.7 ± 11.34 | 2.7 ± 11.34 | 1 | 0.8 ± 5.10 | 5.4 ± 16.40 | 0.005 |
| Leaves | 85.5 ± 22.82 | 90.3 ± 19.86 | 0.124 | 78.8 ± 30.32 | 66.3 ± 35.73 | 0.009 |
| Woody | 0 | 0 | – | 6.8 ± 16.95 | 2.9 ± 10.07 | 0.209 |
| Stones | 0 | 0 | – | 2.3 ± 10.68 | 7.0 ± 17.65 | 0.019 |

less than available (Table 3). On the other hand, scrapes were found more in tall and short forest and less than expected in savanna and secondary forest in El Edén, more in tall forest in Cockscomb (practically the only available habitat type), and more in natural vegetation (Atlantic Forest and Cerrado) and less than expected in edges of vegetation types and plantations in Angatuba (Table 3). In scraping sites, vegetation height was shorter in Angatuba and taller in Cockscomb and El Edén than available, and vegetation cover was higher in Angatuba than available, but according availability in Cockscomb and El Edén (Table 3).

## Size of scrapes

For a total of 202 scrapes from Angatuba, Cockscomb and El Edén width and length were measured. Width and length of the scrapes ranged between 10–36 cm and 14–88 cm, respectively, covering a total surface of 200–2,464 cm$^2$. Width and length were, on average, 21 cm and 25 cm, respectively, this last measure being more variable than the first ($SD = 4.7$ and 10.6, respectively). On average, the ratio length/width was 1.23 ± 0.56 SD (range = 0.53–4.35). However, univariate Kruskall-Wallis tests detected significant differences among study areas, and as a rule, scrapes were shorter and wider in El Edén than in the other two study areas ($H = 30.014$ and 24.574, respectively, $d.f. = 2$, $p < 0.001$ in both cases; Fig. 8). The area where jaguars were absent (Angatuba) presented the intermediate values in scrape sizes (Fig. 8).

It is interesting to note that in Angatuba, where jaguars were absent, variability of data in scrape width was markedly higher than in the other two areas, fact that did not happened for length (Fig. 8). Furthermore, outsider values were mainly observed in Cockscomb for scrape length, and in lesser extent in El Edén, where both jaguars and pumas are present (Fig. 8). Additionally, most of these scrapes with outsider long sizes (approximately >40 cm; Fig. 8) presented higher length/width ratios (mean length/width ratio of outsiders = 2.74 ± 0.90. $n = 13$) than the average (1.26; $n = 201$, $p = 0.04$). Furthermore, in all these 13 scrapes it was not clear the signal of two legs on the ground (Fig. 1). All this information suggests that these scrapes could be made by jaguars and probably with the front legs instead of the hind legs.

**Table 3 Characteristics of the path and vegetation of the sites where scrapes where found (Observed) and availability for each study area (Available).** Statistical tests for comparing the observed and available characteristics are also shown. ND, no data; NA, Not available that vegetation type in the study area; Tall/Atlantic forest, it refers to tall forest for Cockscomb and El Edén, and Atlantic forest for Angatuba; Short forest/cerrado, it refers to short forest for El Edén and cerrado vegetation for Angatuba.

| Variable | Study area | | | | | |
|---|---|---|---|---|---|---|
| | Angatuba | | Cockscomb | | El Edén | |
| | Observed | Available | Observed | Available | Observed | Available |
| **Path width** | | | | | | |
| $N$ | 49 | 268 | 96 | 280 | 70 | 181 |
| Mean $\pm$ SD | $3.0 \pm 0.61$ | $3.5 \pm 1.69$ | $1.7 \pm 0.59$ | $2.5 \pm 2.69$ | $1.7 \pm 0.87$ | $2.3 \pm 0.83$ |
| $t$ student | $t = 1.95, P = 0.052$ | | $t = 2.97, P = 0.003$ | | $t = 5.13, P < 0.001$ | |
| **Path use (%)** | | | | | | |
| $N$ | 51 | 295 | 97 | 280 | 72 | 198 |
| Abandoned | 7.8 | 13.9 | 0 | 0 | 16.7 | 36.4 |
| Rare | 92.2 | 82.4 | 37.1 | 63.6 | 63.9 | 34.8 |
| Frequent | 0 | 3.7 | 62.9 | 36.4 | 19.4 | 28.8 |
| $X^2$ test | $X^2 = 5.02, df = 2, p = 0.081$ | | $X^2 = 19.49, df = 2, p < 0.001$ | | $X^2 = 18.79, df = 2, p < 0.001$ | |
| **Path cleaning (%)** | | | | | | |
| $N$ | | | 97 | 280 | 74 | 197 |
| Yes | ND | ND | 100 | 100 | 90.5 | 75.1 |
| Not | ND | ND | 0 | 0 | 9.5 | 24.9 |
| $X^2$ test | – | | – | | $X^2 = 6.88, df = 1, p = 0.009$ | |
| **Vegetation type (%)** | | | | | | |
| $N$ | 53 | 302 | 97 | 277 | 73 | 201 |
| Savanna | – | NA | – | NA | 1.3 | 6.0 |
| Talk/Atlantic forest | 56.6 | 6.0 | 100 | 98.9 | 13.7 | 1.5 |
| Secundary forest | – | NA | – | NA | 67.1 | 84.6 |
| Short forest/cerrado | 22.6 | 9.3 | – | NA | 17.8 | 8.0 |
| Edge | 18.7 | 52.6 | | 1.1 | – | NA |
| Plantations[a] | 1.9 | 32.1 | – | NA | – | NA |
| $X^2$ test | $X^2 = 126.57, df = 3, p < 0.001$ | | | | $X^2 = 26.15, df = 3, p < 0.001$[b] | |
| **Vegetation height (m)** | | | | | | |
| $N$ | 53 | 264 | 95 | 280 | 73 | 198 |
| Mean $\pm$ SD | $11.2 \pm 3.63$ | $14.1 \pm 4.11$ | $16.7 \pm 3.73$ | $14.7 \pm 2.86$ | $9.8 \pm 3.89$ | $8.5 \pm 3.69$ |
| $t$ student | $t = 4.74, p < 0.001$ | | $t = 5.46, p < 0.001$ | | $t = 2.58, p = 0.010$ | |
| **Vegetation cover (%)** | | | | | | |
| $N$ | 53 | 267 | 97 | 280 | 73 | 198 |
| Mean $\pm$ SD | $3.6 \pm 0.77$ | $2.9 \pm 0.63$ | $3.7 \pm 0.47$ | $3.7 \pm 0.46$ | $3.8 \pm 0.77$ | $3.7 \pm 0.74$ |
| $t$ student | $t = 4.80, p < 0.001$ | | $t = 0.18, p = 0.855$ | | $t = 0.34, p = 0.735$ | |

**Notes.**
[a] It included eucalyptus and pine plantations.
[b] We performed the test despite 20% of the expected values are less than 5 in order to not lose the biological interest of data.
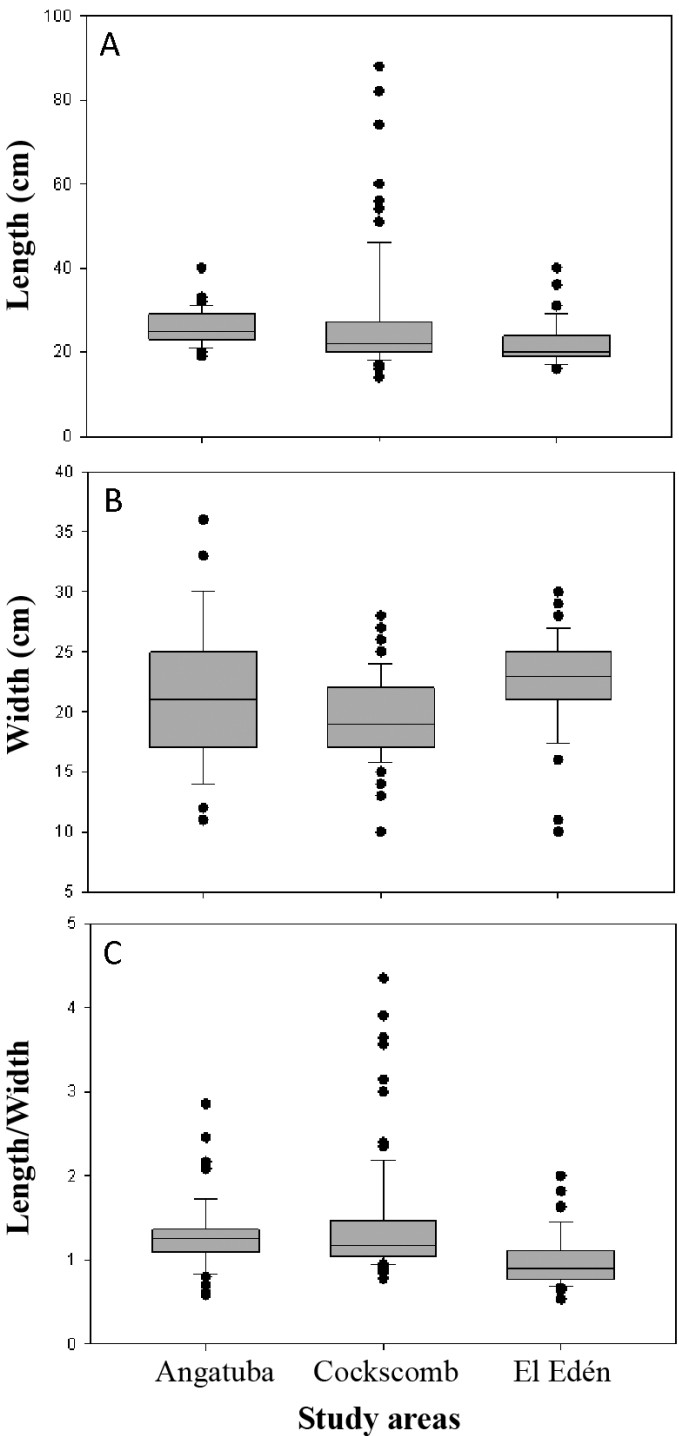

**Figure 8  Length and width of scrapes.** Box plots show differences in length (A), width (B) and the ratio between length and width (C) for scrapes sizes from three study areas. The box indicates the 25th and 75th percentiles, a line within the box marks the median; error bars indicate the 90th and 10th percentiles, and point values outside the last percentiles (outsiders).

**Table 4 Identification of faeces and urine on scrapes.** Felid species and sex determined for faeces or urine (the only one with asterisk) found on scrapes in different study areas. F, female; M, male; ND, sex non-determined.

| Study area | Faeces from scrapes | | Ocelot |
|---|---|---|---|
| | **Jaguar** | **Puma** | |
| Virua | 1F, 1M | 1M | – |
| El Zapotal | – | 1M | – |
| Los Ocotones | 1M, 1ND | – | – |
| La Cojolita | – | 1ND* | – |
| El Edén | 12M | 2M | – |
| Serra das Almas | – | 4M | – |
| Cockscomb | 4M, 1ND | | – |
| Angatuba | | 1M | 1F |

## Scrapes by species and sex, and relative abundance of each species

In 43 scrapes, faeces or urine, respectively, were found over scrapes, and in 32 cases (31 faeces and one urine) the felid could be identified (Table 4). Faeces were from jaguars in 21 scrapes (18 from males, one from female and two undetermined sex) from Viruá, Los Ocotones, El Edén and Cockscomb. Nine faeces were from pumas (all from males) from Viruá, El Zapotal, El Edén, Serra das Almas and Angatuba, and the only urine sample identified come from a puma in La Cojolita (Table 4). One faeces sample was identified as ocelot (Table 4).

As expected in Antaguba and Serra das Almas we did not find faeces from jaguars, finding 26 and 34 faeces from pumas, respectively. In the other three study areas faeces from both species were found with ratios jaguar/puma of 0.4, 1.6 and 11.3 for San Ignacio, El Edén, and Cockscomb, for a total of 40, 50 and 37 faeces collected of both species in each study area, respectively.

## DISCUSSION

### Scraping encounter rates

Scraping was a frequent behaviour in the largest felids of America. Aproximately, one scrape per 1.5 km was found when walking by unpaved track roads or trails in most study areas. Nevertheless, in some areas such as the dry forest of San Ignacio (northern Mexico), scraping behaviour seemed to be rare. This last result is particularly surprising as pumas were more abundant than jaguars and reported in many different areas and habitats (previously and confirmed by this study too), as a felid in which scraping behaviour is frequent (*Harmsen et al., 2010a*; *Logan & Sweanor, 2010*; *Allen, Wittmer & Wilmers, 2014*). In other areas from Amazon basin, we found an apparently low scraping behaviour for any species (F Palomares, pers. obs., 2004 and 2009), while surveying for faeces (*Palomares et al., 2012*; *Palomares et al., 2016*). In other areas such as the Pantanal, jaguars also seem to scrape infrequently (*Schaller & Crawshaw Jr, 1980*). Thus, the factors promoting a noticeable scraping behaviour in the largest felid of America is not related with their

abundance or possible presence of one or both species, and some characteristics of the areas where they live might explain the observed differences.

## Site selection for scraping

It was clear that felids liked to scrape in well delimited and clean paths, independently if these were trails or unpaved track roads, somewhat avoiding sites without trails (but see *Seidensticker et al., 1973*, for a contrasting result for pumas in Idaho). Furthermore, they preferred paths rarely used although not abandoned ones, as the abandoned ones used to have abundant plant regeneration and/or tall grassy vegetation, making, on one hand, it more difficult for felids to move through them, and, on the other hand, to succeed with the objective of signalling the path. Although in some well protected areas, such as Cockscomb, they preferred the more frequently used paths, which are also used for tourist visitation during daylight. Nevertheless, it is interesting to note that this area has a long protection history and tourist visitation, so animals may be relatively used to people, in addition to the chance of encounters with people being smaller due to the crepuscular and nocturnal activity habits of felids in the area (*Harmsen et al., 2010b*). It is also interesting to note that the only two intensive scraping sites detected were at 15–20 m further from the paths. Maybe animals prefer to do these intensive scraping sites slightly further from paths, but it might simply be a consequence of roads or trails being periodically managed and trodden, and this might prevent the accumulation of scrapes. In fact, on paths of one or other type we found a similar number of scrapes close together (what was called clustered scrapes by *Harmsen et al., 2010a*, or community scrapes by *Allen, Wittmer & Wilmers, 2014*). We found no report with such intensive scraping sites as those found in Angatuba in this study.

Scrapes seemed to be also done to signalize other elements of the habitats within territories. In addition to the result obtained on scrape aggregation and re-scraping behaviour, which suggest animals are intentionally marking some given sites and/or habitats, somewhat, scrapes were not made at random and felids selected specific sites to make scrapes both at substrate (microsite) scale and regarding the site level. Selection on microsite scale seemed to be conditioned by the type of substrate that would facilitate signal detection. Felids showed preferences for sites with leaves, and avoided sites with grass and stones, and with a rocky substrate. Stones might make scraping more difficult and less detectable. As a matter of fact, in areas with no rocky substrate, we did not detect selection of any particular microsite.

On the other hand, as a rule, felids did not evenly scrape in all types of vegetation found in the study areas, and the pattern seemed to be also different between study areas according to the main types of vegetation found. In the two more natural areas such as Cockscomb and El Edén, felids selected the more natural and developed vegetation types among available, although they avoided the natural savanna habitat in El Edén. In Angatuba, pumas also selected the natural habitats (Atlantic Forest and Cerrado) and clearly avoided plantations. Therefore, it seems clear felids concentrate scrapes in the best quality areas within their home ranges. This hypothesis is also supported by the linear distribution of scrapes. In the area more altered and heterogeneous (Angatuba), re-scraping behaviour was higher. However, in the more homogeneous area according to habitat quality (Cockscomb),

distribution of scrapes was more uniform and scrapes were mainly situated in regular interval of 100–500 m. Finally, in the two areas where there was some level of regeneration of the natural vegetation (El Eden, and Serra das Almas), patterns of linear distribution of scrapes were similar, with scrapes regularly spaced every 100–500 m or isolated (i.e., scrapes situated >500 m in between). These results suggest that felids mainly concentrate scraping behaviour in the best natural available habitat types, and so scrapes would be related to the use of optimal habitats (e.g., by higher prey availability and/or better refuges against disturbances).

## Scrapes as a communication tool

Our data suggest that scrapes are made with the proposal of communication between individuals. First, the use of well delimited paths for scraping is already pointing in that direction. Felids in general, and jaguar and pumas in particular, like to move on well-established paths (*Harmsen et al., 2010b*; *Palomares et al., 2012*), so they are scraping where there is a higher probability that other individuals or themselves can find the mark. Secondly, they sometimes left faeces or urine on scrapes, and both stuff may inform other individuals about sex, health, and reproductive and social status of individuals (*Allen, Wittmer & Wilmers, 2014*), which reinforced the role of scrapes as a way of intraspecific communication. Third, re-scraping behaviour in some given sites that are visited by several different individuals, suggests this communication role, fact confirmed by *Allen, Wittmer & Wilmers (2014)* and *Allen et al. (2015)* with pumas. Thus, it is also quite probably that intensive scraping sites could be situated in contact borders between territories or to signalize important resources (*Seidensticker et al., 1973*; *Smith, McDougal & Miquelle, 1989*), although our data did not allow testing for this possibility.

## Who scrapes and differences in scrapes and scraping behaviour between jaguar and pumas

Our data suggest that both jaguars and pumas make scrapes and that scraping behaviour in female seems to be a rare event (but see *Allen, Wittmer & Wilmers, 2014*). In fact, we only found jaguar female faeces on one scrape and in some occasions, we have been able to track male pumas (distinguished by footprint size and phototrapping) finding several associated scrapes, whereas it has never been the case when a female was tracked. However, we can be sure that female pumas visited scraping sites as a couple of them were photographed several times in one of the intensive scraping sites detected in the Angatuba study area. This conclusion is in agreement with previous studies (*Seidensticker et al., 1973*; *Logan & Sweanor, 2010*; *Harmsen et al., 2010a*; *Allen, Wittmer & Wilmers, 2014*). However, *Allen, Wittmer & Wilmers (2014)* confirmed that mature females also scrape, although much less frequently than mature male pumas (*Allen et al., 2015*). In addition, we cannot disregard that some of the smaller scrapes found in this study come from ocelots since in other areas not sampled for scrapes in some occasion we have observed small scrapes associated with ocelot tracks (F Palomares, pers. obs., 2004 and 2009).

Our data indicate a human observer cannot distinguish between jaguar and puma scrapes based only on morphology differences. Scrape size of both species widely overlaps.

*Harmsen et al. (2010a)* found similar results in one of our study areas (Cockscomb), although when faeces were over the scrape, they also found that general scrapes with jaguar faeces were larger than scrapes with puma faeces. Despite the large overlap in the size of scrapes of both species, our data indicate that jaguars made longer scrapes with constant width, which were not found in the areas with only pumas (Fig. 8). Such scrapes were often single scrapes as defined by *Harmsen et al. (2010a)*. In fact, we did not find single scrapes in the areas with only pumas as also reported by *Allen, Wittmer & Wilmers (2014)*.

Another aspect that differentiates jaguar and puma scraping behaviour is position on the paths. When only pumas were in an area (Angatuba and Serra das Almas), scrapes were mainly found on the walkable part of the paths (trail centre and wheel tracks on unpaved road), whereas in the areas with jaguars, most scrapes were mainly found in the edge of the path (Cockscomb) or on the edge and centre (El Edén). Note, that in Cockscomb jaguars seem to be much more abundant than pumas (ratio jaguar/puma faeces was 11.3), whereas in El Edén was just slightly higher for jaguars (ratio = 1.6).

Scraping rate was lower in the area where, according to our data on genetic identification of faeces, pumas were less abundant (Cockscomb), and higher in one of the areas with only puma presence (Serra das Almas). Furthermore, in the other area where only pumas were present (Angatuba), scraping rate was not higher than in El Edén (the area with both species present), but this clearly was due to large areas of poor habitat quality (plantations) that were surveyed, where few scrapes were found. However, in this area re-scraping rates were higher than in the other study areas. Thus, it seemed that scraping behaviour was higher in areas with only pumas, and therefore pumas seemed to be the most responsible for scrapes. This result is in agreement with the statements of *Schaller & Crawshaw Jr (1980)*, but it is in opposition to *Harmsen et al. (2010a)*, who suggest that the apparently frequent scraping behaviour of pumas in Cockscomb is probably due to jaguar presence in order to use an inconspicuous way of communication between individuals because of the risk of using a more direct mechanism. If so, they proposed that in areas where pumas were the largest carnivores, scrape rates should be lower. Both in Angatuba and Serra das Almas, where pumas were the largest carnivores, scraping rates were even higher than in Cockscomb. Thus, it seems that a high scraping behaviour in pumas is not related to the presence of jaguars or other larger carnivores.

## CONCLUSION

In some areas, scraping was a frequent behaviour in both jaguars and pumas, and they were mainly made by males, although pumas seemed to be the most responsible for scrapes. Both felids liked to scrape in well delimited and clean paths, independently if these were trails or unpaved track roads, somewhat avoiding sites without trails. Furthermore, they preferred paths rarely used rather than abandoned ones.

Scrapes seemed to be also done to signalize other elements of the habitats within territories since scrapes were not made at random and felids selected specific sites to make scrapes both at the substrate (microsite) scale and regarding the site level. Nevertheless, selection on the microsite scale seemed to be conditioned by the type or substrate rather

than to facilitate signal detection. Felids selected the more natural and developed vegetation types among those available to scrape, and clearly avoided plantations. Thus, scrapes were more patchily distributed in the areas more altered and were more evenly distributed in the more natural areas.

Scrapes from jaguars and pumas can not be distinguished based only in morphology differences, except for some long scrapes (>40 cm) that are made by jaguars. The main aspect differentiating jaguar and puma scrapes was the position on the paths, with pumas mostly scraping on the walkable part of the paths (trail centre and wheel tracks on unpaved road), whereas jaguars scraped on the edge of the path, and only jaguars made single scrapes.

## ACKNOWLEDGEMENTS

We thank the managers of the Edén Ecological Reserve (Marco Lazcano), El Zapotal Ecological Reserve (Pronatura Península de Yucatán: Juan Carlos Faller and María Andrade), Fazendas Três Lagoas (Denise Carmignani), Arca (Caetano Carmignani), Estação Ecológica de Angatuba (Bárbara Heliodora Prado), Serra das Almas Natural Heritage Reserve (Thiago Roberto Soares Vieira) for their logistical support. C Zapata and two anonymous referees made useful comments on the manuscript.

### Funding

This study was supported by the project CGL2010-16902 of the Spanish Ministry of Research and Innovation, the project CGL2013-46026-P of Ministerio de Economía, Industria y Competitividad, and the excellence project RNM 2300 of the Junta de Andalucía and the Formación de Profesorado Universitario fellowship AP2010-5373 from the Spanish Ministry of Education. Luciano M. Verdade has a productivity scholarship from Ministério da Ciência, Tecnologia, Inovações e Comunicações (Proc. No. 309468/2011-6). Marina Zanin is supported by Desenvolvimento Científico e Tecnológico fellowship number 312627/2015-7. The funders had no role in study design, data collection and analysis, decision to publish, or preparation of the manuscript.

### Grant Disclosures

The following grant information was disclosed by the authors:
Spanish Ministry of Research and Innovation: CGL2010-16902.
Ministerio de Economía, Industria y Competitividad: CGL2013-46026-P.
Junta de Andalucía and the Formación de Profesorado Universitario fellowship: RNM 2300.
Spanish Ministry of Education: AP2010-5373.
Ministério da Ciência, Tecnologia, Inovações e Comunicações: 309468/2011-6.
Desenvolvimento Científico e Tecnológico: 312627/2015-7.

## Competing Interests

The authors declare there are no competing interests.

## Author Contributions

- Francisco Palomares conceived and designed the experiments, performed the experiments, analyzed the data, contributed reagents/materials/analysis tools, prepared figures and/or tables, authored or reviewed drafts of the paper, approved the final draft.
- Noa González-Borrajo conceived and designed the experiments, performed the experiments, analyzed the data, prepared figures and/or tables, authored or reviewed drafts of the paper, approved the final draft.
- Cuauhtémoc Chávez, Yamel Rubio and Bart Harmsen performed the experiments, contributed reagents/materials/analysis tools, authored or reviewed drafts of the paper, approved the final draft.
- Luciano M. Verdade conceived and designed the experiments, contributed reagents/materials/analysis tools, authored or reviewed drafts of the paper, approved the final draft.
- Rocio Monsa performed the experiments, authored or reviewed drafts of the paper, approved the final draft.
- Begoña Adrados and Marina Zanin analyzed the data, authored or reviewed drafts of the paper, approved the final draft.

## Field Study Permissions

The following information was supplied relating to field study approvals (i.e., approving body and any reference numbers):

Sampling in Brazil was carried out under licenses no. 11214 and no. 13781 of ICMBio, and no 131/2005 CGFAU/LIC, 13883-1 SISBIO and 15664-1 SISBIO of the Instituto Brasileiro do Meio Ambiente—IBAMA, and at the Mexican sites under the license SGPA/DGVS/549 of the Dirección General de Vida Silvestre (Semarnat). Sampling in Belize and posterior exportation of feces was approved by the Forest Department of Belize with the Ref. No. CD/60/3/14 (25). Fecal samples were exported from Brazil to Spain for genetic analysis under IBAMA/CGEN Autorização de Acesso license no 063/05 and IBAMA/CITES export licenses no 0123242BR, 08BR002056/DF and 09BR003006/DF, and from Mexico to Spain under the export licenses no MX33790 and MX42916 from the Secretaria de Medio Ambiente/CITES.

## Data Availability

The raw data are provided in the Supplemental Files.

## Supplemental Information

Supplemental information for this article can be found online at http://dx.doi.org/10.7717/peerj.4983#supplemental-information.

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
