# Peer review of "Scraping marking behaviour of the largest Neotropical felids"

_PeerJ, doi:10.7717/peerj.4983_

## Round 0.1 · original submission · Minor Revisions

You will find below some very helpful comments and suggestions that would help to improve the quality and impact of the manuscript. Thanks to the reviewers, and I look forward to receiving a revised manuscript.

Reviewer 1 ·

Basic reporting

The manuscript “Scraping marking behaviour of the largest Neotropical felids” is a welcome contribution to the better understanding of scrape-marking behaviour of jaguars and pumas. The document is well written, but the text should be entirely revised to improve its quality:
- standardization of the name of biomes (e.g. Atlantic Forest and Cerrado, always with capital letters - proper nouns);
- the word “Reserve” must be with capital letter when is part of the name of certain study area (the same applies to other words in proper nouns);
- standardization of study area names (e.g. the authors use Reserva Natural Serra das Almas and Serra das Almas Nature Reserve, but the correct is Serra das Almas Private Reserve of Natural Heritage);
- the correct name is Viruá (Viruá National Park) and not Virua;
- review the appropriate use of comma or semicolon;
- see other adjustments (e.g. jaguars are much more abundant “than” pumas).

The Figures 2 and 6 should be improved (standardization of information as described in the text - Figure 2; and improve resolution – both figures).

Some figures are not correctly associated with the text in which they were referenced (e.g. page 21, line 493). Review all.

The information in the Supplementary Material (tables) should be translated into English (some cells are in Spanish). Likewise, date of records needs to be adjusted.

The raw data regarding the scrapes associated with faeces or urine and that had the identification of the feline made by DNA should be duly indicated from the other data in the table (Supplementary Material).

Experimental design

See "Validity of the findings" for methodological issues.

Validity of the findings

Abstract - The authors cited “a total of 349 felid scrapes along 467 km of paths surveyed, obtaining a finding rate of 0.560 scrapes per km”. However, this rate is related to other values. Review the text and use the appropriate values.

Study areas - The authors cite five areas and list four locations. At this point, I suggest considering four locations. In addition, other study areas are cited in the item "Felid relative abundance and assigning scrapes to species and sex", but they should be presented together with the main sampling areas. This is reinforced because the complementary areas are also presented in Figure 2. The legend of Figure 2 also needs to be revised.

Surveying for scrapes - The types of path surveyed (car tracks, trails and cross country) should be presented in this item.

Felid relative abundance and assigning scrapes to species and sex – The authors inform they “cannot be sure about the faeces or urine found over scrapes necessarily had to come from the species and individuals who made the scrape”. This issue needs to be better addressed. Is there previous information (from the authors or references) on the habit of felines demarcating on marks left by other species or individuals?

Data analysis – (1) General analysis strategies should be presented in this item and not only in the results. (2) According the authors: “A site was defined as an area where one or more scrapes were found within a given distance. To determine this distance, we used the maximum distance we found (21 m) between scrapes in two intensive scrapping sites that we found in the Angatuba area”. Would that maximum distance be biased? Considering that there are peculiarities in the different areas studied and that the marking behaviour also varies between areas (see data of the authors themselves), the authors' proposal needs to be better justified. The same must be done for the other distances (lineal distribution of scrapes). What was the criterion for defining the distances? (3) Similarly, the authors must to better justify the width of 22 m to adapt the Ripley´s function.

Results – (1) The authors found “269 scrapes of felids, which meant a scraping rate of 0.560 scrapes per km surveyed”. Inform the sampling distance considered in the calculation of the rate (if they sampled 467 km, the rate is wrong - it should be 0.576). (2) Inform the degree of freedom in all the results of statistical tests. (3) The results of chi-square test for differences among study areas (distance of scrap sites) and the results of linear distribution of scrapes should be informed in the text (page 14, lines 323 to 331). In the first case, I suggest using paired analyzes to compare the study areas and the different distance categories. (4) The presentation of the data needs to be standardized (e.g. page 16: length/width ratios = “mean ratio of outsiders = 2.74 ± 0.90, n = 13”, and the average = “1.26; n = 201, se = 0.04”).

Discussion – (1) According the authors, they “cannot discard that some of the smaller scrapes found in this study come from ocelots since in some occasion we have observed ocelot scrapes after tracking them in sandy soils”. The scraping marking behaviour by ocelots needs to be better addressed. Is there information on the size of ocelots’ scrapes? (2) Authors interpret that the “selection on microsite scale seemed to be conditioned by the type of substrate rather than to facilitating signal detection”. However, the avoidance of sites with grass and stones, or with rocky substrate (less “mouldable” than sand or clay soil) indicates to me the selection of microsites that facilitate the detection and maintains the marking for longer. The authors themselves consider that “stones might make scraping more difficult and less detectable”. Review these issues. (3) Author cited that “in Cockscomb jaguars seem to be much more abundant than pumas (ratio jaguar/puma faeces was 11.3), whereas in El Edén was just slightly higher for jaguars (ratio= 1.6)”. Are there population abundance estimates for these areas? Can the scrap marking rate be used as a direct indicator of abundance? Justify.

Conclusion - Add that the jaguar scrapes were also single scrapes (not only long scrapes).

·

Basic reporting

No comment

Experimental design

No comment

Validity of the findings

No comment

Additional comments

Dear Authors,

This contribution increases the knowledge about scrapping behavior of the biggest Neotropical felids, the jaguar (Panthera onca) and the puma (Puma concolor). The emphasis is on the comparison of scrapes characteristics and scrapping behavior of both pumas and jaguars in areas differing in the density of these species, and areas where only pumas were present. The working hypothesis of the manuscript is that both felids use scrapes for marking proposals, as well as to communicate with other individuals, and to signalize particular points or areas of interest.
I liked reading the manuscript, and enjoyed the figures of scraps and the video of the puma scrapping. I have little concerns and suggestions, and I indicate them in the list below:

Methods
1) Surveying for scrapes
#130-#134: The surveys were conducted between five days and two months, but it is not clear to me if the same path was surveyed once or twice or more times in one period (except for Serra das Almas). Could you give some details?
2) Selection of scrapping sites
#175: Why did you chose 5m as the distance between the 1x1m plot where the scrap is located and a 1x1 plot on the same path to test for selection of microsites? What did you rely on when selecting that distance (5m)? Could you briefly explain it?

Results
3) Field samplings and finding rate
#236: you say you “…surveyed 467 km of paths once or more times……” and found a total of 349 scrapes. Later in #238, you wrote that you find 269 scrapes. In Table 1 overall kilometers surveyed are 467 and total number of scrapes found in the five study areas 269. So, could you explain this disparity?
Also #237 and #238: I find this sentence a little confusing in terms on the duration of the sampling periods and as I wrote above in “1) Surveying for scrapes”, it is not clear how many times the different paths were surveyed.
4) Position of scrapes in paths
#254: here you mention a total of 318 scraps including those far away from path from Angatuba. But the number is different from the total presented above (n = 349).
5) Re-marking rate and linear distribution of scrapes on paths
# 282: here is clear that you removed the 2 scrapes from San Ignacio from analysis (it is only a comment).
6) Size of scrapes
# 331 and #332: I think you mean that the length/width ratio of scrapes is more similar between Angatuba and Cockscomb than among these last sites and El Edén.

Discussion
7) Site selection for scrapping
# 391 to #394. Why do you think felids avoid abandoned paths for scrapping (or prefer not abandoned ones)? And in Cockscomb, why do you think felids prefer the more frequently used paths for scrapping? could you say something about this?
# 394: The last sentence is confusing: “ …so animals may be use to people using the path during daylight, when…” and need to be rewritten

Figures
Figure 2: you may also include the total number of scrapes found in each study area as a number in the figures. Please replace Belize by Cockscomb in the figure.
Figure 3: please include “(ISA1 and ISA2)” after “….intensive scrapping points…”in the text of the figure.
Figure 7: please, replace “….on linear paths for all data altogether and for every study area” by “…on linear paths for each study area and for overall data”


References

Check the entire section, there are a lot of references that are not cited in the text (e.g. #511, #524, #534, #582, #591, #595, #605, #611, #614, #622).


Best wishes,

Sonia Zapata

Reviewer 3 ·

Basic reporting

The manuscript entitled is overall well written and brings interesting information to the audience of PeerJ and increases the knowledge of predators’ marking ecology.
The structure of the manuscript and background information seems appropriate, as well as the amount and quality of figures.
Overall, the manuscript has some minor points that can be improved or corrected.
Please, see below.

Experimental design

The sampling design an analysis seems appropriate although there might be some source of bias. The magnitude of this bias might be minor or not but it would be important that addressed in the discussion by the authors.

Validity of the findings

The authors present an important amount of data often rare to find on this topic.

Additional comments

## Here is a list of minor comments ordered by lines, followed by general comments mostly.

Line 71 – According to bibliography list the citation to Harmsen et al. 2010 should be 2010a or 2010b. This is repeated later in text (e.g lines 73, 364)
Line 73 – Remove the semicolon after Harmsen et al.
Line 88 - Ordering the hypothesis by scale would be easier to understand and a rough measure of size would be welcomed.
Lina 99 - Reserva Natural Serra das Almas should be 5 (4 is repeated).
Line 208 - Probably want to use “linear” instead of “lineal”, here and in the following lines.
Line 119 – “Private” instead of “privative”
Line 213 – “belonging” instead of “bellowing”
Line 221 – I suggest to add “off” before “areas”
Line – I would not say similar in most occasions, actually are quite different in two-thirds of the sites (Cockstaomb and Serra Almas), I would rather say that differences were more pronounced in Car tracks
Line 295 – provide the tests results.
Line 361 - If by “north Mexico” you want to refer to San Ignacio site, please specify otherwise provide a citation.
Line 364 - “Harmsen” instead of “Harnsem”
Map - Mention why San Ignacio is not displayed in more detail as the other sites. Explain the meaning of the different types of symbols. Use “Cockscomb” instead of “Belize”, as in text.
Bibliography:
Some of the species names are not in italics.
In the Baddley and Turner 2005 citation, the name of Turner was written and only put the initial for last name.
Many citations are not mentioned in the text, some are in the following list:
Schenkel R. 1947. Expression studies of wolves. Behaviour 1: 1-129.
Vila, C., Urios V. and Castroviejo J. 1994. Use of faeces for scent marking in Iberian wolves
(Canis lupus). Canadian Journal of Zoology 72: 374-377.
Sunquist M, Sunquist F. 2002. Wild Cats of the World. University of Chicago Press, Chicago,
USA.

Sillero‐Zubiri, C. and Macdonald D. W. 1998. Scent‐marking and territorial behaviour of Ethiopian wolves Canis simensis. Journal of Zoology 245: 351-361.
Kitchener, A. 1991. The natural history of the wild cats. The Natural history of mammals series.
Comstock Publishing Association, Ithaca, New York.
Kohn, M. H. and Wayne R. K. 1997. Facts from feces revisited. Trends in ecology &
evolution, 12: 223-227.
Menotti-Raymond M., David V. A., Lyons L. A., Schäffer A. A., Tomlin J. F., Hutton M. K.,
and O'Brien S. J.. 1999. A genetic linkage map of microsatellites in the domestic cat (Felis
catus). Genomics, 57: 9-23.
Molteno A. J., Sliwa A., and Richardson P. R. K. 1998. The role of scent marking in a free-
ranging, female black-footed cat (Felis nigripes). Journal of Zoology 245: 35-41.
Pal, S. K. 2003. Urine marking by free-ranging dogs (Canis familiaris) in relation to sex, season,
place and posture. Applied Animal Behaviour Science 80: 45-59.
Peters, G., and Hast M. H. 1994. Hyoid structure, laryngeal anatomy, and vocalization in felids
(Mammalia: Carnivora; Felidae). Zeitschrift für Säugetierkunde, 59: 87-104.
Pilgrim, K. L., McKelvey K. S., Riddle A. E., and Schwartz M. K. 2005. Felid sex identification
based on noninvasive genetic samples. Molecular Ecology Notes 5: 60-61

General comments:
In the discussion, you start talking about “who scrapes” and then discuss about scrape characteristics (disregarding species) and then in the last section you go back to differences between jaguar and puma. I would suggest merging the sections where you compare species, and using more order similar to results.
If you are acknowledging that jaguar and puma differ in their likely to leave a scrape, an estimation of abundance ratio based on droppings on scrapes is most likely biased based on your conclusions. Given that is not an important topic on the discussion I would not include relative abundance estimations on the manuscript. If you like to include it, a discussion on this possible bias would be an important addition.
In the part of the manuscript on felid species discriminated by sex, the sample size for pumas is smaller than jaguars this might explain why you do not detect any female puma, also the number of female scraping jaguars is very low. I would say that scraping in females is rare in both species; I think a more general conclusion will be more suitable for the data set.
Probably a scrape in the middle of the forest is less likely to find that one on a trail. Additionally, your sampling technique was to search for scrapes along trails and the scraping sites out of the trails were found following tracks, you did not perform searching in the forests either account for differential detection probability. Harmsen et al (2010b) found that jaguars are more willing to use forest matrix (instead trails) than pumas, this might be influencing the conclusion that pumas scrape more often than jaguars (is a possibility that there are lots of hard to find jaguar scrapes in the forest). Since you sampled on trails only, making the conclusion that pumas scrape more than jaguars relative to markings on trails might be more accurate coming from your data. Although most authors agreed that pumas scrape more that jaguars, quantitative data is rare, for that a discussion regarding the possible bias on scrape detection would be appreciated.

---

## Round 0.2 · accepted · Accept

I am a Section Editor for this area, and I have been asked by the journal office to replace your previous Academic Editor as he is unavailable to make the decision on your submission.

I have reviewed your submission materials, the reviews and your responses, and I am happy to suggest acceptance. It is clear you have made the corrections required. This looks like an interesting paper.

Reviewer 3 ·

Basic reporting

The manuscript entitled is overall well written and brings interesting information to the audience of PeerJ and increases the knowledge of predators’ marking ecology.
The structure of the manuscript and background information seems appropriate, as well as the amount and quality of figures.
Overall, the authors adressed the minor concerns mentioned by revisors.

Experimental design

The sampling design an analysis seems appropriate although there might be some source of bias (mentioned in my previous revision).
The magnitude of this bias might be minor or not but it would be important that addressed in the discussion by the authors and besides I already expressed my concern the authors do not acknowledge this possible source of bias that they are not accounting for.

Validity of the findings

The authors present an important amount of data often rare to find on this topic.

Additional comments

I think the changes in the new version of the manuscript covers all my previous concerns.